# Coupling RNN with LLM: Does Their Integration Improve Highly Order-Sensitive Language Understanding?

## Abstract

Pretrained large language models (LLMs) have demonstrated remarkable success across various language modeling tasks. However, in domain-specific applications, particularly those involving highly order-sensitive data, general LLMs exhibit limitations in achieving state-of-the-art performance. One notable issue is that the contextual embeddings by LLMs still lack a strong positional inductive bias, especially for long and highly ordered sequences, leading to the *lost in the middle* problem. In this work, we utilized the potential of sequential models (RNNs) with LLMs to address the issue and investigate whether RNN integration improves LLM performance. The LLM generates rich contextual embeddings using the attention mechanism of the Transformer. The RNN further processes the LLM embeddings to capture the contextual semantics of long and order-sensitive dependencies. The LLM-RNN model leverages the potential of both Transformer and recurrent structures to enhance performance in domain-specific tasks. We perform a wide range of experiments leveraging multiple types of LLMs (encoder-only, encoder-decoder, and decoder-only) and RNNs (GRU, LSTM, BiGRU, and BiLSTM) across diverse public and real-world datasets to investigate the potential (either positive or negative) of LLM-RNN models. The experimental results highlight the superiority of the LLM-RNN model, showing improvements in commonsense reasoning, code understanding, and biomedical reasoning tasks.

## 1 Introduction

Large Language Models (LLMs) achieved groundbreaking performance in diverse NLP tasks, including commonsense analysis Cai et al. (2024); Chang et al. (2024), code understanding Du et al. (2024), code summarization and generation Yan et al. (2024); Riddell et al. (2024), biomedical text retrieval Xu et al. (2024), question answering Robinson & Wingate (2023), and text summarization, generation, and translation Tu et al. (2024); Papi et al. (2024); He et al. (2024). For LLMs, training data is a pivotal factor leading to enhanced model performance Wei et al. (2022b), and at the same time, incorporating larger training data has substantially increased model size. However, these models are laying the foundation toward artificial general intelligence Bubeck et al. (2023). Thus, LLMs have attained continuous attention from both academia Wei et al. (2022a); Zhao et al. (2023) and industry Anil et al. (2023); Achiam et al. (2023).

Considering the wide range of use and success of LLMs in recent years, numerous methodologies and techniques have been developed to adapt these general-purpose models to domain-specific downstream tasks. Beyond the conventional model fine-tuning approach, where all parameters are adjusted during training Howard & Ruder (2018), which is expensive, prompt-based adaptation methods have been introduced to modulate the behavior of frozen LLMs using carefully designed prompts Li & Liang (2021); Tian et al. (2024); Brown et al. (2020); Lester et al. (2021). Low-rank adaptation techniques allow the pretrained model weights to remain fixed while introducing trainable rank-decomposition matrices, significantly reducing the number of trainable parameters Hu et al. (2021). Rather than modifying the core parameters of LLMs, these approaches freeze the pretrained weights and typically introduce additional trainable parameters. In addition, various innovations in LLMs, such as incorporating knowledge graph representations of text, feature fusion (e.g., late or early), sequential model integration, and adjustments of layer architecture, are being explored

to enhance the structural and functional capabilities of LLMs Mostafizer et al. (2021); Bugueño & de Melo (2023); Rahman et al. (2025).

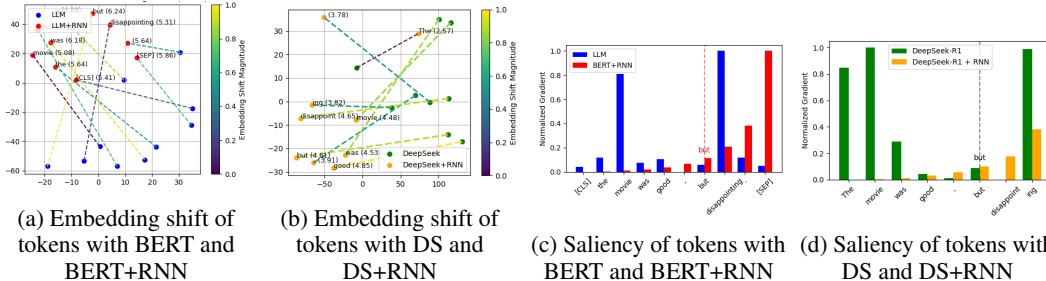

(a) Embedding shift of tokens with BERT and BERT+RNN

(b) Embedding shift of tokens with DS and DS+RNN

(c) Saliency of tokens with BERT and BERT+RNN

(d) Saliency of tokens with DS and DS+RNN

Figure 1: A token-level comparison of embedding shifts and saliency values for the sentence *"The movie was good, but disappointing"*. BERT and DeepSeek serve as the base LLMs, each coupled with a BiGRU RNN.

In recent years, LLMs have achieved great success in tackling real-world applications (e.g., translation, generation, and summarization), and the adaptability of LLMs to various downstream tasks has increased significantly. However, there is still room for performance improvement as they continue to show limitations in capturing and providing fundamental knowledge Pan et al. (2024); Lewis et al. (2020). Lexical and semantic diversity, order sensitivity, the presence of long dependencies, unfamiliar symbols and words in text, and imbalanced datasets pose continuous challenges for LLMs, especially in sentiment analysis and code understanding Chang et al. (2024); Rahman et al. (2025). LLM-generated codes are helpful for programmers, but often lack clarity and maintainability, making it difficult for programmers in terms of debugging, extensibility, and maintenance Liang et al. (2024); Vaithilingam et al. (2022), especially for novice programmers. In the study Poria et al., 2020, the authors explored significant challenges and highlighted some interesting research directions. Another notable issue is that contextual embeddings produced by LLMs lack a strong positional inductive bias, especially in long, highly ordered sequences, leading to phenomena such as the *lost in the middle* problem Wu et al. (2025). These limitations point to the need for models that can integrate richer contextual and sequential cues, particularly for highly order-sensitive data in domain-specific applications. As illustrated in Figure 1, a token-level comparison of embedding shifts and saliency values for the sentence *"The movie was good, but disappointing"* is analyzed using BERT and DeepSeek (DS) as the LLMs, and BiGRU as the recurrent neural networks (RNN). The integration of RNN with BERT and DS significantly enhances context and order sensitivity, as evidenced by the model's ability to down-weight the token *good* and emphasize the contrastive cue *but*, thereby correctly prioritizing *disappointing* as the dominant token. The observed embedding shifts and saliency adjustments indicate that the LLM-RNN captures nuanced, order-sensitive dependencies that standard transformer-based models may underrepresent. Motivated by these observations, we pose the following research question:

**Coupling RNN with LLM: Does Their Integration Improve Highly Order-Sensitive Language Understanding?**

To address the research question, we comprehensively examine the integration of RNNs with domain-specific pretrained LLMs, encompassing encoder-decoder models (CodeT5 Wang et al. (2021), CodeT5$^{+}$ Wang et al. (2023)), encoder-only models (RoBERTa Liu et al. (2019), BioLinkBERT Yasunaga et al. (2022), CodeBERT Feng et al. (2020)), and decoder-only models (GPT2 Radford et al. (2019), DeepSeek-Coder Guo et al. (2024), DeepSeek-R1 Guo et al. (2025)), in order to evaluate LLM-RNNs' performance on *commonsense reasoning*, *biomedical reasoning*, and *code understanding* tasks. We leverage RNN variants (e.g., GRU, LSTM, BiGRU, and BiLSTM) and also perform comprehensive hyperparameter tuning for model optimization. The proposed LLM-RNN model combines the strengths of both Transformer and Recurrent architectures. The LLM serves as the primary encoder, tokenizing and transforming input sequences into meaningful embeddings. The LLM-generated embeddings are passed through a dropout layer to mitigate overfitting before being processed with the RNN. The RNN captures sequential dependencies in the text, enhancing the model's ability to understand structural, order, and logical relationships. Finally, an FC layer maps the RNN outputs to target class labels, enabling the classification layer to perform the downstream predictive tasks.

We conducted extensive experiments on multiple public datasets involving order sensitivity, with a particular focus on coding datasets across three diverse tasks. To achieve optimal performance, we fine-tuned the model hyperparameters. Our findings demonstrate that coupling RNN with LLM enables the model to better capture context, leading to significant improvements in performance. Figure 2 presents the averaged accuracy comparison between the LLM-RNN and stand-alone LLM across the three tasks using multiple benchmark datasets. Notably, LLM-RNN achieves accuracy (Avg.) improvements of approximately **+1.11%**, **+3.81%**, and **+0.37%** for commonsense reasoning, code understanding, and biomedical reasoning, respectively, compared to stand-alone LLM models. The obtained results highlight the effectiveness of the proposed approach, particularly in handling highly order-sensitive data such as source code. In summary, the key contributions are:

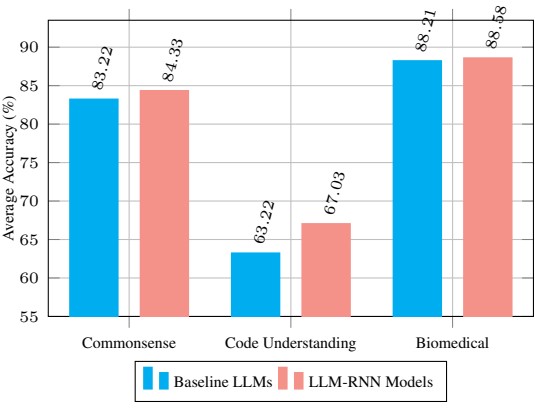

Figure 2: Comparison of average accuracy between LLM-RNN models and stand-alone counterparts across three tasks, evaluated on multiple benchmark datasets.

- To the best of our knowledge, this work presents the first comprehensive evaluation of coupling RNN with LLM, including encoder-only, encoder-decoder, and decoder-only architectures, across multiple publicly available datasets and diverse downstream tasks. We systematically investigate the performance of RNN-coupled LLMs within each architectural category across datasets.

- We integrate RNN architectures with LLMs and fine-tune the hyperparameters to harness the complementary strengths of both Transformer-based models and the sequential learning capabilities of RNNs. Since LLMs generate rich, contextually relevant token embeddings, RNNs further refine the contextual representations by capturing the structural, temporal, and order-sensitive dependencies inherent in the input. Furthermore, we adjust the layers (e.g., dropout, activation, and FC ) of the LLM-RNN framework for optimal results.

## 2 METHODOLOGY

In this section, we describe the coupling of RNN with LLM to create LLM-RNN model. LLM-RNN model combines the strengths of the Transformer and RNN architectures to improve efficiency and accuracy in downstream tasks. Figure 3 shows the framework of the LLM-RNN model.

### 2.1 CONTEXTUAL EMBEDDING WITH LLM

The tokenization strategies, such as Byte Pair Encoding (BPE), WordPiece, byte-level BPE, and SentencePiece in LLM, are not fixed but vary from model to model; tokenization is primarily used to represent input with the aim of reducing out-of-vocabulary problems Sathe et al. (2025); Sennrich et al. (2016). Let $S = \{s_1, s_2, \ldots, s_n\}$ be the input (e.g., text or code), and the tokenization of the input $S$ can be written as $G = \text{Tokenizer}(S) = \{g_1, g_2, \ldots, g_n\}$, where $G$ is the sequence of tokens. Now, each token $g_i$ is mapped to three (03) key categories, namely input ID (id$_i \in \mathbb{Z}$), token ID (gg$_i \in \{0, 1\}$), and attention mask (m$_i \in \{0, 1\}$), which will be fed into the pre-trained LLM encoder. Since we include encoder-only, decoder-only, and encoder-decoder LLMs as part of our

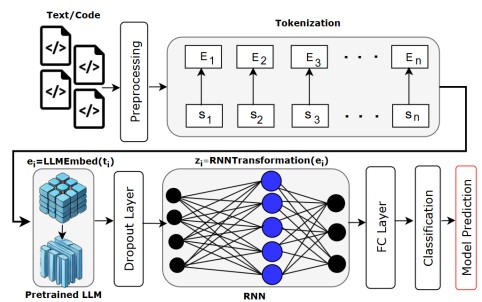

Figure 3: Architectural framework of the RNN-coupled LLM

model and investigation, they adopted different objectives; for example, encoder-only and encoder-decoder models typically use span-masking (denoising) losses, and decoder-only models use causal language modeling (next-token prediction). For our tasks (e.g., classification), all types of LLMs act as relevant encoders, and their final hidden representations are mapped to task-specific labels. Using a Transformer architecture, token embeddings $E(G') = \{e_1, e_2, \ldots, e_n\}$ are derived, where $e_i = \text{LLMEmbed}(g_i)$. These embeddings are processed by the Transformer encoder to produce contextual representations $H(G') = \text{TransformerEncoder}(E(G')) = \{h_1, h_2, \ldots, h_n\}$, with $h_i$ being the contextual embedding for $g_i$. In our approach, the contextual embeddings are reprocessed through the RNN, which captures sequential dependencies for downstream tasks such as reasoning and code defect detection.

## 2.2 Contextual Embedding Reprocessing with RNN Sequential Model

The LLM-RNN highlights the effectiveness of RNN models in capturing rich contextual details, establishing them as a popular choice for sequential data analysis tasks due to their enhanced performance and resilience. The output embeddings from the final layer $L$ of the LLM model are represented as a sequence $H^{(L)} = \{h_1^{(L)}, h_2^{(L)}, \ldots, h_n^{(L)}\}$, where $h_i^{(L)} \in \mathbb{R}^d$ denotes the $i$-th embedding in the sequence, and $d$ is the dimensionality of the embeddings. To reduce overfitting, a dropout operation is applied to these embeddings, resulting in $h_i^{\text{drop}} = \text{Dropout}(h_i^{(L)})$, where $h_i^{\text{drop}} \in \mathbb{R}^d$. To align the dimensionality of the LLM output embeddings with the input requirements of the RNN, a linear transformation is applied to each embedding: $z_i = W_{\text{linear}} h_i^{\text{drop}} + b_{\text{linear}}$, where $z_i \in \mathbb{R}^{d_{\text{RNN}}}$, $W_{\text{linear}} \in \mathbb{R}^{d_{\text{RNN}} \times d}$ is the weight matrix, and $b_{\text{linear}} \in \mathbb{R}^{d_{\text{RNN}}}$ is the bias vector. The sequence of transformed embeddings $\{z_1, z_2, \ldots, z_n\}$ is then processed by the RNN, which computes the hidden states sequentially: $h_i^{\text{RNN}} = \text{RNN}(z_i, h_{i-1}^{\text{RNN}})$, where $h_i^{\text{RNN}} \in \mathbb{R}^{d_{\text{RNN}}}$ is the $i$-th hidden state, and $h_{i-1}^{\text{RNN}}$ is the hidden state from the previous time step. The final output sequence of the RNN is given by $H_{\text{RNN}} = \{h_1^{\text{RNN}}, h_2^{\text{RNN}}, \ldots, h_n^{\text{RNN}}\}$, which combines the contextual embeddings from LLM with the sequential dependencies modeled by the RNN to enhance the semantic understanding and order-sensitivity. The sequential recurrence of the LLM-RNN approach allows the model to explicitly learn and track token-order dependencies. The proposed LLM-RNN framework can be written as $G \xrightarrow{\text{Tokenizer}} G' \xrightarrow{\text{Embedding}} E(G') \xrightarrow{\text{Transformer Encoder}} H(G') \xrightarrow{\text{RNN + Classifier}} \hat{y}$. Further explanation is available in the Appendix A.

## 2.3 FC and Classification Layers

A dropout layer is applied to the RNN output $H_{\text{RNN}}$ to mitigate overfitting, resulting in $H' = \text{Dropout}(H_{\text{RNN}})$, followed by an FC layer that maps the RNN hidden states to class logits: $\text{logits}_i = W_o H' + b_o$. Unlike traditional approaches that apply a softmax activation to produce class probabilities, we used raw logits directly.

## 3 Experimental Setup

### 3.1 Hyperparameters

The performance of LLMs is highly dependent on selecting appropriate hyperparameters. In this work, we conducted extensive experiments with various hyperparameter configurations to evaluate model performance on multiple tasks. Appendix B details the hyperparameters used for fine-tuning during model training. For BiLSTM and BiGRU architectures, the number of RNN hidden units ($h$) is doubled ($2 \times h$) due to their bidirectional processing capabilities, which incorporate both forward ($\overrightarrow{h}$) and backward ($\overleftarrow{h}$) information. During training, categorical cross-entropy is employed to calculate the loss, defined as: $\mathbf{L}(g) = -\sum_{j=1}^{K} u_j \, log(\bar{u}_j)$. Where $g$ and $K$ represent the model parameter and the number of classes, respectively, while $u_j$ and $\bar{u}_j$ denote the true and predicted labels for the $j^{th}$ sample.

## 3.2 METRICS

The performance of the models is evaluated using standard metrics Rahman et al. (2025); Younas et al. (2022), including accuracy (A), precision (P), recall (R), and F1-score (F1). The accuracy metric (A) is defined as: $A = \frac{1}{N} \sum_{l=1}^{|K|} \sum_{x:f(x)=l} B(f(x) = \hat{f}(x))$. Where, $B$ is a function that returns 1 if the predicted class is correct and 0, otherwise. $K$ represents the total number of classes, and $f(x) \in K = \{1, 2, 3, \cdots\}$. In addition to $\mathbf{A}$, weighted-precision ($P_\psi$), recall ($R_\psi$), and F1-score ($F1_\psi$) are computed to provide an unbiased and comprehensive performance evaluation.

## 3.3 DATASETS

We evaluated the LLM-RNN approach using five public and three real-world datasets across multiple tasks. For commonsense reasoning, we employed the IMDb, Twitter US Airline, and Sentiment140 datasets. The IMDb dataset Maas et al. (2011) comprises 50,000 reviews evenly split between positive and negative sentiments, providing a balanced dataset with 50% of samples in each class. The Twitter US Airline dataset Tan et al. (2022) contains 14,640 tweets categorized into three sentiment classes: positive, neutral, and negative. The Sentiment140 dataset Go et al. (2009) is a substantial collection of approximately 1.6 million tweets curated by Stanford University in 2009 for sentiment analysis. This dataset is equally balanced, with 50% of tweets representing positive sentiment and 50% representing negative sentiment. For code understanding, we used the defect detection, SearchAlg, SearchSortAlg, and SearchSortGT datasets. The defect detection benchmark dataset, sourced from CodeXGLUE Zhou et al. (2019), is utilized to assess the model's ability to identify code defects. The other three datasets—SearchAlg, SearchSortAlg, and SearchSortGT—are collected from AOJ Rahman et al. (2024), a reputed repository of real-world source code. Finally, the NCBI dataset O'Leary et al. (2024) is leveraged for the biomedical reasoning task. Section *Order-sensitivity of Coding Data* in Appendix D further discusses why coding data is highly order-sensitive.

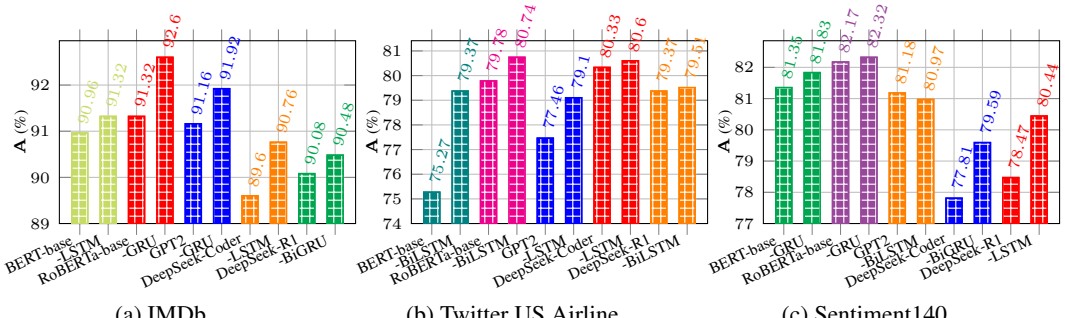

|                | (a) IMDb | (b) Twitter US Airline | (c) Sentiment140 |

Figure 4: The best $\mathbf{A}$ scores achieved by the BERT-base, RoBERTa-base, GPT2, DeepSeek-Coder, and DeepSeek-R1, along with their corresponding RNN-coupled models.

## 4 RESULTS AND ANALYSIS

We conducted extensive experiments using various LLM-RNN models. Figure 4 presents the $\mathbf{A}$ scores of the best-performing BERT-, GPT2-, DeepSeek-Coder-, DeepSeek-R1-, and RoBERTa-RNN[1] models, alongside their corresponding base counterparts, for the commonsense reasoning task on IMBd, Twitter, and Sentiment140 datasets. Figure 4a demonstrates that the BERT-base model achieved an $\mathbf{A}$ score of approximately 90.96%, while the BERT-LSTM model attained 91.32%, reflecting a 0.36% improvement. The RoBERTa-GRU model attained an $\mathbf{A}$ score of 92.60%, marking a 1.28% improvement compared to the RoBERTa-base model. Among the decoder-only models, GPT2-GRU, DeepSeek-R1-BiGRU, and DeepSeek-Coder-LSTM exhibited improvements of 0.76%, 1.16%, and 0.40%, respectively, compared to their corresponding base models. Similar trends are observed in the Twitter and Sentiment140 datasets, as depicted in Figures 4b and 4c, respectively. In most cases, integrating RNNs with both encoder-only and decoder-only models enhanced the performance of LLMs. Figure 4 highlights that the RoBERTa-RNN models achieved greater improvements than the other models across all datasets. Additionally, Table 1 presents the

---

[1]RoBERTa-RNN encompasses four models, each integrating a different RNN variant: LSTM, GRU, BiLSTM, and BiGRU. This similarly applies to the BERT-, CodeBERT-, GPT2-, DeepSeek-R1-, DeepSeek-Coder-, BioLinkBERT-, CodeT5-, and CodeT5$^+$-RNN models.

best weighted scores ($\mathbf{F1}_\psi$, $\mathbf{P}_\psi$, and $\mathbf{R}_\psi$). The results obtained clearly demonstrate that integrating RNNs improves the performance (as marked by ↑) of both encoder-only and decoder-only LLMs on commonsense reasoning tasks. A detailed breakdown of the results is provided in Table 6 in the Appendix.

Table 1: The best results for commonsense reasoning obtained using encoder-only and decoder-only LLM architectures, including their respective base models and RNN-coupled variants. **Note:** Notation such as BERT + LSTM (1) indicates that the model achieved the best performance on the IMDb dataset; this notation applies similarly to the other models and datasets.

| Model (Base and RNNs) | IMDb (1) | | | Twitter Airline (2) | | | Sentiment140 (3) | | |
|---|---|---|---|---|---|---|---|---|---|
| | $\mathbf{F1}_\psi$ | $\mathbf{P}_\psi$ | $\mathbf{R}_\psi$ | $\mathbf{F1}_\psi$ | $\mathbf{P}_\psi$ | $\mathbf{R}_\psi$ | $\mathbf{F1}_\psi$ | $\mathbf{P}_\psi$ | $\mathbf{R}_\psi$ |
| BERT | 90.96 | 90.96 | 90.96 | 75.88 | 76.62 | 75.27 | 81.31 | 81.56 | 81.35 |
| + LSTM (1), BiLSTM(2), GRU(3) | 91.32↑ | 91.35 | 91.32 | 78.18↑ | 78.01 | 78.42 | 81.83↑ | 81.84 | 81.83 |
| RoBERTa | 91.31 | 91.44 | 91.32 | 80.12 | 80.70 | 79.78 | 82.17 | 82.21 | 82.17 |
| + BiLSTM(1), GRU(2,3) | 92.96↑ | 92.96 | 92.96 | 80.93↑ | 81.47 | 80.60 | 82.32↑ | 82.32 | 82.32 |
| GPT2 | 91.16 | 91.19 | 91.16 | 76.92 | 77.80 | 77.46 | 81.18 | 81.19 | 81.18 |
| + GRU(1), LSTM(2), BiLSTM(3) | 91.92↑ | 91.94 | 91.92 | 78.86↑ | 78.74 | 79.10 | 80.97 | 80.97 | 80.97 |
| DeepSeek-R1 | 90.09 | 90.21 | 90.08 | 79.36 | 79.42 | 79.37 | 78.47 | 78.47 | 78.47 |
| + BiGRU(1), BiLSTM(2), LSTM(3) | 90.49↑ | 90.52 | 90.48 | 79.03 | 78.96 | 79.51 | 80.43↑ | 80.53 | 80.44 |
| DeepSeek-Coder | 89.59 | 89.92 | 89.60 | 79.89 | 79.80 | 80.33 | 77.81 | 77.81 | 77.81 |
| + LSTM(1,2), BiGRU(3) | 90.76↑ | 90.77 | 90.76 | 80.14↑ | 80.18 | 80.60 | 79.57↑ | 79.68 | 79.59 |

For the code understanding task, we employed encoder-only, decoder-only, and encoder-decoder LLMs and conducted extensive experiments under various hyperparameter settings. Table 2 presents a comparative analysis of the best $\mathbf{A}$ and F1 scores achieved by the top-performing models alongside their corresponding base models on the defect detection benchmark dataset. The $\mathbf{A}$ scores for the base models—RoBERTa, CodeBERT, DeepSeek-Coder, CodeT5, and CodeT5$^+$—are 61.05%, 62.08%, 65.37%, 64.86%, and 64.90%, respectively. In comparison, their RNN-augmented counterparts (except for DeepSeek-Coder) demonstrated notable performance gains. Specifically, the RoBERTa-BiGRU model achieved an $\mathbf{A}$ score of 66.40%, indicating a 5.35% improvement over the base RoBERTa. Similarly, the CodeBERT-GRU model reached 66.03%, showing a 3.95% improvement. The CodeT5-GRU and CodeT5$^+$-BiGRU models attained $\mathbf{A}$ scores of 67.90% and 67.79%, corresponding to gains of 3.04% and 2.89%, respectively, over their base models. Among all models, CodeT5-GRU recorded the highest $\mathbf{A}$ score. More detailed results are provided in Table 7 in the Appendix. These results highlight the effectiveness of integrating RNN architectures with LLMs, demonstrating significant enhancements in performance for code understanding tasks.

Table 2: Comparison of the best $\mathbf{A}$ and F1 scores achieved by the top-performing encoder-only, decoder-only, and encoder-decoder LLMs integrated with RNNs, along with their respective base models, on the defect detection dataset.

| Model | | Learning Rate ($l$) | Optimizer ($\triangle$) | Hidden Units ($h$) | A (%) | F1 (%) | |
|---|---|---|---|---|---|---|---|
| LLM | RNN | | | | | Weighted ($\psi$) | Macro ($\mu$) |
| RoBERTa | - | - | - | - | 61.05 | - | - |
| | BiGRU | $1e^{-5}$ | NAdam | 512 | 66.40 (↑) | 64.76 | 64.0 |
| CodeBERT | - | - | - | - | 62.08 | - | - |
| | GRU | $2e^{-5}$ | AdamW | 512 | 66.03 (↑) | 65.32 | 65.0 |
| DeepSeek-Coder | - | $1e^{-5}$ | AdamW | 256 | 65.37 | 64.08 | 63.39 |
| | BiLSTM | $1e^{-5}$ | AdamW | 256 | 59.11 | 58.86 | 58.42 |
| CodeT5 | - | - | - | - | 64.86 | 64.74 | - |
| | GRU | $1e^{-4}$ | AdamW | 512 | 67.90 (↑) | 67.18 | 67.0 |
| CodeT5$^+$ | - | - | - | - | 64.90 | 64.74 | - |
| | BiGRU | $2e^{-5}$ | RMSProp | 256 | 67.79 (↑) | 66.82 | 66.0 |

To further evaluate the impact of integrating RNN with LLM, we conducted experiments on three real-world coding datasets. Figure 5 presents a comparative analysis of the $\mathbf{A}$ scores across RoBERTa, CodeBERT, CodeT5, CodeT5$^+$, and DeepSeek-Coder, both in their stand-alone and RNN-coupled forms. Among the models, CodeT5$^+$-BiLSTM achieved the highest $\mathbf{A}$ scores of approximately

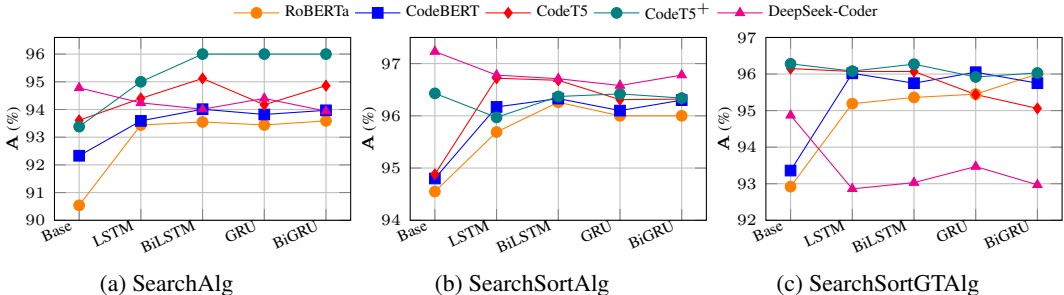

(a) SearchAlg  (b) SearchSortAlg  (c) SearchSortGTAlg

Figure 5: Comparison of $A$ scores for the top-performing RoBERTa-RNN, CodeBERT-RNN, CodeT5-RNN, CodeT5+-RNN, and DeepSeek-Coder LLMs with their respective base models on the SearchAlg, SearchSortAlg, and SearchSortGTAlg datasets.

Table 3: Performance comparison between the base LLMs and their best-performing RNN-coupled counterparts in terms of weighted $\mathbf{F1}_\psi$, $\mathbf{P}_\psi$, and $\mathbf{R}_\psi$ scores. The evaluation includes encoder-only, decoder-only, and encoder-decoder LLMs, each integrated with different RNN variants.

| Model | SearchAlg (4) | | | SearchSortAlg (5) | | | SearchSortGTAlg (6) | | |
|---|---|---|---|---|---|---|---|---|---|
| | $\mathbf{F1}_\psi$ | $\mathbf{P}_\psi$ | $\mathbf{R}_\psi$ | $\mathbf{F1}_\psi$ | $\mathbf{P}_\psi$ | $\mathbf{R}_\psi$ | $\mathbf{F1}_\psi$ | $\mathbf{P}_\psi$ | $\mathbf{R}_\psi$ |
| RoBERTa | 90.53 | 90.55 | 90.54 | 94.40 | 94.51 | 94.55 | 92.79 | 92.89 | 92.92 |
| + BiGRU (4, 6) BiLSTM (5) | 93.63↑ | 93.90 | 93.59 | 96.25↑ | 96.25 | 96.26 | 96.00↑ | 96.10 | 96.00 |
| CodeBERT | 92.36 | 92.40 | 92.33 | 94.73 | 94.93 | 94.80 | 93.22 | 93.37 | 93.36 |
| + BiLSTM(4, 5), LSTM(6) | 94.04↑ | 94.11 | 94.01 | 96.34↑ | 96.37 | 96.33 | 96.04↑ | 96.21 | 96.02 |
| CodeT5 | 93.63 | 93.67 | 93.61 | 94.88 | 95.14 | 94.88 | 96.01 | 96.03 | 96.15 |
| + BiLSTM(4, 6), LSTM(5) | 95.12↑ | 95.15 | 95.12 | 96.72↑ | 96.76 | 96.72 | 96.01↑ | 96.06 | 96.07 |
| CodeT5$^+$ | 93.38 | 93.39 | 93.38 | 96.42 | 96.44 | 96.43 | 96.26 | 96.32 | 96.28 |
| + BiGRU(4), GRU(5), BiLSTM(6) | 94.42↑ | 94.43 | 94.42 | 96.42↑ | 96.44 | 96.42 | 96.26 | 96.31 | 96.27 |
| DeepSeek-Coder | 94.79 | 94.81 | 94.78 | 97.23 | 97.23 | 97.23 | 94.78 | 94.84 | 94.87 |
| + GRU(4), LSTM(5, 6) | 94.42 | 94.50 | 94.40 | 96.77 | 96.78 | 96.78 | 92.97 | 92.90 | 92.86 |

96.00% and 96.27% on the SearchAlg and SearchSortGTAlg datasets, respectively. Additionally, DeepSeek-Coder-BiGRU attained an $\mathbf{A}$ score of 96.78% on the SearchSortAlg dataset. These results indicate that coupling RNN with LLM generally leads to significant performance improvements. However, this trend does not consistently hold for DeepSeek-Coder, where RNN integration offered limited or no gain. We also computed weighted evaluation metrics—$\mathbf{F1}_\psi$, $\mathbf{P}_\psi$, and $\mathbf{R}_\psi$—to provide a more nuanced assessment. The results, summarized in Table 3, reveal consistent performance improvement patterns across all datasets, further validating the benefit of RNN integration. Additional detailed results and descriptions are provided in Table 8 in the Appendix.

Figure 6 demonstrates that incorporating RNN with LLM enhances performance on biomedical reasoning tasks. The RoBERTa-LSTM model achieved an absolute improvement of 0.60% in $\mathbf{A}$ score over the RoBERTa-base model, which attained 88.15%. BioLinkBERT-GRU yielded a modest gain of 0.14%, while GPT2-LSTM and DeepSeek-R1-LSTM achieved increases of 0.10% and 0.53%, respectively, compared to their corresponding base models. Table 4 presents detailed weighted evaluation metrics to offer a comprehensive assessment of model performance. All RNN-augmented models outperformed their base counterparts in terms of $\mathbf{F1}_\psi$ score. Additional detailed results are provided in Table 9 in the Appendix. These findings highlight that coupling RNN with LLM can lead to consistent and significant gains in biomedical reasoning accuracy and robustness.

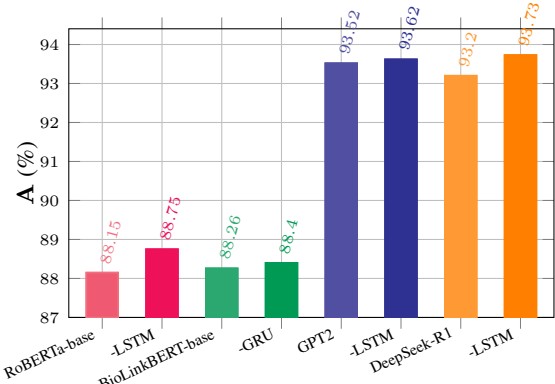

Figure 6: Comparison of the best $\mathbf{A}$ score achieved by the base models—RoBERTa, BioLinkBERT, GPT-2, and DeepSeek-R1—and their RNN-coupled variants on the NCBI dataset.

### 4.1 HYPERPARAMETER SENSITIVITY

We conducted a sensitivity analysis focusing on key hyperparameters: learning rates, optimizers, and the number of RNN hidden units. Figure 7a illustrates that the RoBERTa-BiLSTM model achieved optimal performance on the Twitter dataset with a learning rate of $l = 1e^{-5}$ and hidden units $h = 256$, outperforming other parameter configurations. For the Sentiment140 dataset, the RoBERTa-GRU model failed to achieve optimal results

Table 4: The best results for biomedical reasoning using RoBERTa, BioLinkBERT, GPT2, and DeepSeek-R1 models on the NCBI dataset.

| Model | $\mathbf{F1}_\psi$ | $\mathbf{P}_\psi$ | $\mathbf{R}_\psi$ |
|---|---|---|---|
| RoBERTa | 86.75 | 85.40 | 88.15 |
| + LSTM | 87.02 (↑) | 85.37 | 88.75 |
| BioLinkBERT | 86.81 | 85.43 | 88.26 |
| + GRU | 86.86 (↑) | 85.39 | 88.40 |
| GPT2 | 93.52 | 93.52 | 93.52 |
| + LSTM | 93.63 (↑) | 93.64 | 93.62 |
| DeepSeek-R1 | 93.20 | 93.20 | 93.20 |
| + LSTM | 93.75 (↑) | 93.98 | 93.73 |

with a $l = 1e^{-4}$ and $h = 256$, as shown in Figure 7b. These findings suggest that a lower $l$ significantly enhances the model's performance, and highlight the importance of selecting appropriate parameters for optimal results. Additionally, Figure 8 compares the $\mathbf{A}$ scores obtained using the two top-performing optimizers, $\Delta = $ AdamW, NAdam. The results indicate that the models consistently achieved superior performance across most configurations, with the sole exception occurring at a learning rate of $l = 1e^{-6}$ for both optimizers. The model performance with these optimizers is sensitive to the $l$ values, and excessively lowering $l$ decreases the performance gains.

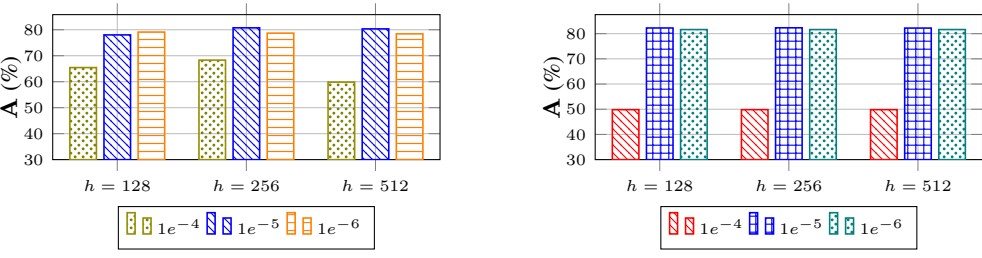

(a) RoBERTa-BiLSTM on Twitter dataset      (b) RoBERTa-GRU on Sentiment140 dataset

Figure 7: Impact of hyperparameters on model performance.

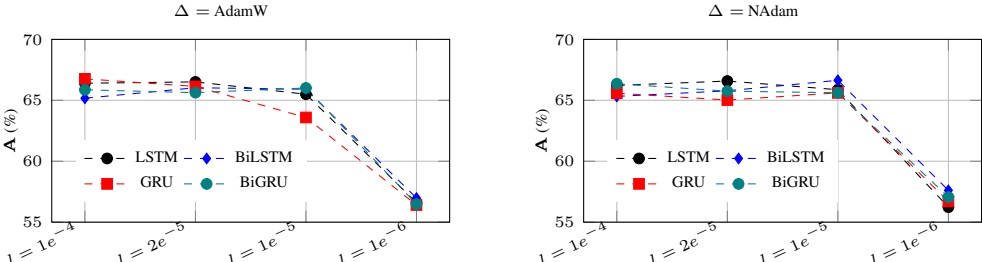

Figure 8: Accuracy ($\mathbf{A}$) scores of CodeT5-RNN models with several key parameters on defect detection dataset.

### 4.2 ANALYSIS OF TRAINING PARAMETERS AND TIME

Figure 9 shows the training parameters and training times for the top-performing LLM-RNN models and their base counterparts. Notably, BiGRU coupled LLMs exhibit the highest number of training parameters. Among these, the RoBERTa and CodeBERT models each comprise approximately 125 million parameters, while the BioLinkBERT, CodeT5 and CodeT5$^+$ models contain around 112 million parameters, as depicted in Figure 9a. In addition, the trainable parameter counts for decoder-only models are also calculated: DeepSeek-Coder, DeepSeek, and GPT-2 contain approximately 1.28 billion, 1.55 billion, and 124.44 million parameters, respectively. Interestingly, despite having fewer parameters, the CodeT5 and CodeT5$^+$ models, when combined with RNN variants, required significantly more training time compared to the RoBERTa and CodeBERT models, as shown in Figure 9b. This discrepancy can be attributed to architectural features, variations in tokenization strategies, potentially less efficient computational optimization when coupling RNNs with CodeT5 and CodeT5$^+$, and specific hyperparameter settings, all of which may collectively contribute to the extended training time.

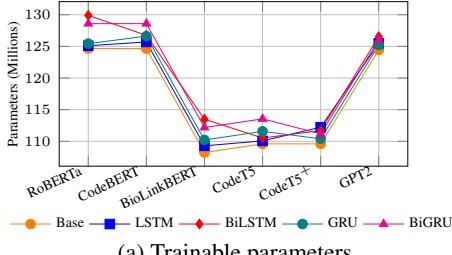 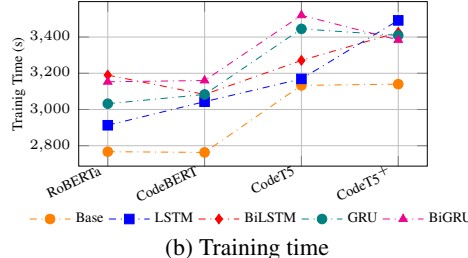

(a) Trainable parameters                      (b) Training time

Figure 9: Overview of trainable parameters and training times for the RNN-augmented LLMs and their counterparts.

## 5 DISCUSSION

The contextual embeddings generated by the LLM are further refined through an RNN layer, enabling the model to better capture deeper semantic structures and effectively handle highly order-sensitive data. Our experiments investigate how coupling RNNs with LLMs enhances semantic understanding and improves performance on tasks that require sensitivity to input order. The results show that the LLM-RNN models yield notable performance gains: an average accuracy improvement of approximately +1.11% for commonsense reasoning, +0.37% for biomedical reasoning, and a significant +3.81% for code understanding tasks. These findings underscore the utility of RNNs in augmenting LLMs, particularly for domains such as source code, where the sequential structure and order of the input are critical. We conducted an additional ablation study to examine how LLM-RNN performance varies when the RNN is attached to early, intermediate, or final-layer embeddings of the LLM. The results, presented in Appendix H, show that attaching the RNN to the LLM's final-layer embeddings provides the strongest performance. Overall, the LLM-RNN integration proves especially effective in modeling highly order-sensitive data and contributes to mitigating the well-known *lost-in-the-middle* problem associated with Transformer models.

## 6 CONCLUSION

Our study encompasses three distinct application domains—commonsense reasoning, code understanding, and biomedical reasoning—spanning both benchmark and real-world datasets. We systematically explore encoder-only, decoder-only, and encoder-decoder LLM architectures, each coupled with four RNN variants (LSTM, BiLSTM, GRU, and BiGRU), thereby offering one of the most extensive evaluations to date of LLM-RNN hybrid models. To the best of our knowledge, no prior work has simultaneously considered this breadth of architectural configurations, LLM and RNN types, and task diversity. Our results consistently show that the sequential modeling capacity of RNNs complements the contextual representation of LLMs, particularly for order-sensitive inputs. Among all evaluated tasks, code understanding emerges as the most sensitive to sequential dependencies. In this context, LLM-RNN models achieved an average accuracy improvement of approximately 3.81% over their stand-alone LLM counterparts. It underscores the effectiveness of RNNs in refining LLM-generated embeddings by capturing syntactic and structural nuances inherent to programming languages. Similarly, for commonsense and biomedical reasoning tasks, the integration of RNNs led to measurable performance gains. These results confirm that RNNs enhance the expressiveness of LLM embeddings not only in structurally rigid domains like code but also in semantically rich and noisy domains such as social media and biomedical. Overall, coupling RNNs introduces beneficial inductive biases for modeling temporal and sequential structures, an area that Transformer-based LLMs may underutilize. The LLM-RNN framework thus offers a compelling direction for enhancing the contextual understanding of LLMs, especially in tasks where the order of information is critical. Our findings affirmatively answer the research question posed in this study: *Yes, coupling RNN with LLM enhances model performance across a range of downstream tasks, particularly in scenarios requiring strong order-sensitive language understanding.*

### USE OF THE LARGE LANGUAGE MODELS (LLMS)

In this study, LLMs (generative models) were used exclusively for language polishing. Their use was limited to refining the wording in the introduction, methodology, stylizing equations, a few sentences in results, and conclusion. No LLMs were employed in conducting experiments, analyzing results, or drawing conclusions.

## ETHICS STATEMENT

This work adheres to the ICLR Code of Ethics by proposing a novel method that advances research on integrating RNNs with LLMs for highly order-sensitive language understanding. All experiments were conducted using publicly available datasets, and no human subjects, private data, or personally identifiable information were involved. We evaluated the proposed framework on three distinct reasoning tasks—biomedical, commonsense, and code understanding—to demonstrate its effectiveness. The results highlight the potential of the framework to provide meaningful benefits for advancing language understanding.

## REPRODUCIBILITY STATEMENT

All experiments were run with fixed random seeds, and dataset preprocessing steps, training configurations, and evaluation metrics are fully described in the paper and Appendix. To support reproducibility, we will release the complete source code (including Jupyter notebooks, Python scripts, trained checkpoints, and evaluation results), along with the curated dataset and a README for detailed guidance, either during the rebuttal phase upon request or with the camera-ready submission.

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

APPENDIX

## A  LLM-RNN FOR ORDER-SENSITIVE LANGUAGE UNDERSTANDING

Let the input be a sequence of raw tokens: $S = \{s_1, s_2, \ldots, s_n\}$. The LLM tokenizes this sequence (e.g., using Byte Pair Encoding) into subword tokens: $G = \text{Tokenizer}(S) = \{g_1, g_2, \ldots, g_m\}$. Each token $g_i$ is embedded using the LLM's embedding layer: $e_i = \text{LLMEmbed}(g_i) \in \mathbb{R}^d$. The Transformer encoder then generates contextualized embeddings:

$$h_i = \text{TransformerEncoder}(e_1, \ldots, e_m)_i \in \mathbb{R}^d.$$

Although these embeddings $h_i$ capture contextual information, transformer attention mechanisms do not enforce a strong positional inductive bias—particularly in long sequences. This limitation often results in phenomena such as being *"lost in the middle"* Wu et al. (2025).

To introduce stronger order modeling, these contextual embeddings are passed through a recurrent neural network (RNN). First, the embeddings $\{h_1, \ldots, h_m\}$ are projected to match the input dimensionality $d_{\text{RNN}}$ required by the RNN:

$$z_i = W_{\text{linear}} \cdot h_i + b_{\text{linear}}, \quad z_i \in \mathbb{R}^{d_{\text{RNN}}}.$$

Next, the projected embeddings are processed by the RNN:

$$h_i^{(R)} = \text{RNN}(z_i, h_{i-1}^{(R)}).$$

In bidirectional RNNs (e.g., BiGRU), the forward ($\overrightarrow{h_i}$) and backward ($\overleftarrow{h_i}$) hidden states are concatenated:

$$h_i^{(R)} = \left[ \overrightarrow{h_i}; \overleftarrow{h_i} \right].$$

This sequential recurrence allows the model to explicitly learn and track token-order dependencies, in contrast to the transformer's permutation-invariant attention.

Transformers leverage positional encodings and global self-attention, which are effective for capturing context but insufficient for enforcing strict sequential constraints. RNNs, on the other hand, compute representations step-by-step, naturally modeling precedence and temporal causality. Bidirectional RNNs enhance this by capturing both past and future dependencies, which is especially useful for disambiguating contrastive or hierarchical structures.

### A.1  CONTRASTIVE SENTIMENT DISAMBIGUATION

Consider the sentence: *"The movie was good, but disappointing."* A transformer-based LLM may assign similar attention weights to *good* and *disappointing* due to their strong semantic features. However, the RNN layer processes the sentence sequentially, identifying *but* as a contrastive cue. As a result, it down-weights *good* and emphasizes *disappointing*, correctly identifying the negative sentiment. The LLM-RNN framework combines rich contextualization with robust sequence modeling, resulting in improved performance in domains where token order is critical (e.g., source code, clinical narratives, and contrastive statements). This hybrid architecture introduces a temporal inductive bias that is especially beneficial for modeling structured, logical, and dependent language patterns.

## B  HYPERPARAMETERS

Optimal hyperparameter configurations are crucial for maximizing model performance. Identifying these optimal settings typically requires extensive combinatorial tuning. Prior to reporting our final results, we conducted comprehensive hyperparameter optimization for each task. Our experiments involved fine-tuning approximately ten LLMs—spanning encoder-only, decoder-only, and encoder-decoder architectures—combined with four different RNN variants, three optimizers, multiple dropout rates, learning rates, and RNN hidden unit sizes. The specific values used for tuning are summarized in Table 5.

Table 5: The list of hyperparameters for the experiments

| Parameter-Name | Values |
|---|---|
| Pre-trained LLMs | BERT, RoBERTa, Code-BERT, BioLinkBERT, CodeT5, CodeT5$^+$, GPT2, DeepSeek-Coder and -R1 |
| RNNs | LSTM, BiLSTM, GRU, BiGRU |
| Optimizer ($\Delta$) | AdamW, NAdam, RMSprop |
| Loss function ($\mathcal{L}$) | Categorical Cross Entropy (cross_entropy) |
| Epochs ($epoch$) | 5 |
| Dropout ($d$) | 0.1, 0.2 |
| Learning rates ($l$) | $1e^4, 1e^5, 2e^5, 1e^6$ |
| Hidden units ($h$) of RNNs | 128, 256, 512 |

## C IMPLEMENTATION DETAILS

The experiments are conducted on a system running RHEL 8.8. The hardware configuration included an Intel Ice Lake (Xeon Platinum 8358) (2 sockets *32 cores/socket), 256 GB of RAM, and an NVIDIA A100 80GB PCIe graphics card.

Table 6: Quantitative results for commonsense reasoning obtained using encoder-only and decoder-only LLM architectures, including their respective base models and various RNN-coupled variants. The encoder-only models comprise BERT and RoBERTa, while the decoder-only models include GPT2, DeepSeek-R1, DeepSeek-Coder, and LLama-3-8B-Instruct. All models are trained for 5 epochs on the IMDb, Twitter US Airline, and Sentiment140 datasets.

| Model | IMDb | | | Twitter US Airline | | | Sentiment140 | | |
|---|---|---|---|---|---|---|---|---|---|
| | $\mathbf{F1}_\psi$ | $\mathbf{P}_\psi$ | $\mathbf{R}_\psi$ | $\mathbf{F1}_\psi$ | $\mathbf{P}_\psi$ | $\mathbf{R}_\psi$ | $\mathbf{F1}_\psi$ | $\mathbf{P}_\psi$ | $\mathbf{R}_\psi$ |
| BERT-Base | 90.96 | 90.96 | 90.96 | 75.88 | 76.62 | 75.27 | 81.31 | 81.56 | 81.35 |
| -GRU | 91.12 | 91.14 | 91.12 | 77.57 | 77.45 | 77.73 | 81.83↑ | 81.84 | 81.83 |
| -LSTM | 91.32↑ | 91.35 | 91.32 | 77.72 | 77.54 | 78.01 | 81.75 | 81.75 | 81.75 |
| -BiLSTM | 91.24 | 91.25 | 91.24 | 78.18↑ | 78.01 | 78.42 | 81.81 | 81.81 | 81.81 |
| RoBERTa-Base | 91.31 | 91.44 | 91.32 | 80.12 | 80.70 | 79.78 | 82.17 | 82.21 | 82.17 |
| -GRU | 92.60 | 92.64 | 92.60 | 80.93↑ | 81.47 | 80.60 | 82.32↑ | 82.32 | 82.32 |
| -LSTM | 92.08 | 92.08 | 92.08 | 80.32 | 80.47 | 80.33 | 82.29 | 82.29 | 82.29 |
| -BiLSTM | 92.96↑ | 92.96 | 92.96 | 80.73 | 80.94 | 80.74 | 82.25 | 82.25 | 82.25 |
| GPT2 | 91.16 | 91.19 | 91.16 | 76.92 | 77.80 | 77.46 | 81.18 | 81.19 | 81.18 |
| -GRU | 91.92↑ | 91.94 | 91.92 | 78.24 | 78.11 | 78.55 | 80.92 | 80.93 | 80.92 |
| -BiGRU | 90.83 | 90.90 | 90.84 | 78.42 | 78.61 | 78.28 | 80.96 | 80.96 | 80.96 |
| -LSTM | 91.52 | 91.56 | 91.52 | 78.86↑ | 78.74 | 79.10 | 80.87 | 80.87 | 80.87 |
| -BiLSTM | 91.44 | 91.50 | 91.44 | 77.84 | 77.67 | 78.14 | 80.97 | 80.97 | 80.97 |
| DeepSeek-R1 | 90.09 | 90.21 | 90.08 | 79.36 | 79.42 | 79.37 | 78.47 | 78.47 | 78.47 |
| -GRU | 89.76 | 89.77 | 89.76 | 78.74 | 78.67 | 78.83 | 80.24 | 80.33 | 80.26 |
| -BiGRU | 90.49↑ | 90.52 | 90.48 | 78.33 | 78.18 | 78.55 | 80.21 | 80.33 | 80.23 |
| -LSTM | 89.69 | 89.87 | 89.68 | 78.62 | 78.60 | 78.83 | 80.43↑ | 80.53 | 80.44 |
| -BiLSTM | 89.26 | 89.36 | 89.28 | 79.03 | 78.96 | 79.51 | 80.24 | 80.35 | 80.26 |
| DeepSeek-Coder | 89.59 | 89.92 | 89.60 | 79.89 | 79.80 | 80.33 | 77.81 | 77.81 | 77.81 |
| -GRU | 90.12 | 90.12 | 90.12 | 79.06 | 79.14 | 79.64 | 79.47 | 79.53 | 79.48 |
| -BiGRU | 90.08 | 90.14 | 90.08 | 79.51 | 79.59 | 80.05 | 79.57↑ | 79.68 | 79.59 |
| -LSTM | 90.76↑ | 90.77 | 90.76 | 80.14↑ | 80.18 | 80.60 | 79.28 | 79.40 | 79.30 |
| -BiLSTM | 89.84 | 89.93 | 89.84 | 78.45 | 78.28 | 78.83 | 79.47 | 79.51 | 79.48 |
| LLama-3-8B | 81.15 | 81.18 | 81.16 | 69.15 | 68.71 | 69.81 | 72.82 | 72.83 | 72.82 |
| -LSTM | 86.03↑ | 86.10 | 86.04 | 71.63↑ | 72.47 | 73.50 | 75.12↑ | 75.15 | 75.13 |
| -BiLSTM | 85.99 | 86.08 | 86.00 | 71.16 | 71.29 | 72.68 | 73.98 | 74.27 | 74.05 |

## D  ORDER-SENSITIVITY OF CODING DATA

Coding data is inherently more order-sensitive than social media or biomedical text due to the strict syntactic and semantic constraints of programming languages. In source code, the position and sequence of tokens such as keywords, variables, operators, and delimiters directly determine the program's logic and functionality. Even minor changes in token order can lead to syntactic errors, unintended behaviors, or entirely different computational outcomes. In contrast, social media and biomedical texts, while often semantically complex, exhibit greater tolerance for word reordering or paraphrasing without significantly altering meaning. As a result, models processing code must precisely capture long-range and hierarchical dependencies to understand control flow, nesting, and execution order, making code understanding tasks especially sensitive to token sequencing.

Table 7: Comparison of **A** and F1 scores between top-performing models (RoBERTa-RNN, CodeBERT-RNN, CodeT5-RNN, DeepSeek-Coder-RNN, and CodeT5$^+$-RNN) and their base models on the defect detection dataset.

| Model | | Learning Rate ($l$) | Optimizer ($\Delta$) | Hidden Units ($h$) | A (%) | F1 (%) | |
|---|---|---|---|---|---|---|---|
| LLM | RNN | | | | | Weighted ($\psi$) | Macro ($\mu$) |
| RoBERTa | - | - | - | - | 61.05 | - | - |
| CodeBERT | - | - | - | - | 62.08 | - | - |
| CodeT5-Small | - | - | - | - | 63.40 | - | - |
| CodeT5-Base | - | - | - | - | 64.86 | 64.74 | - |
| CodeT5$^+$ | - | - | - | - | 64.90 | 64.74 | - |
| DeepSeek-Coder | - | $1e^{-5}$ | AdamW | 256 | 65.37 | 64.08 | 63.39 |
| RoBERTa | BiGRU | $1e^{-5}$ | NAdam | 512 | 66.40 | 64.76 | 64.0 |
| CodeBERT | GRU | $2e^{-5}$ | AdamW | 512 | 66.03 | 65.32 | 65.0 |
| CodeT5 | GRU | $1e^{-4}$ | AdamW | 512 | 67.90 | 67.18 | 67.0 |
| CodeT5$^+$ | BiGRU | $2e^{-5}$ | RMSProp | 256 | 67.79 | 66.82 | 66.0 |
| DeepSeek-Coder | LSTM | $1e^{-5}$ | AdamW | 256 | 53.00 | 48.07 | 46.57 |
| DeepSeek-Coder | BiLSTM | $1e^{-5}$ | AdamW | 256 | 59.11 | 58.86 | 58.42 |
| DeepSeek-Coder | GRU | $1e^{-5}$ | AdamW | 256 | 56.95 | 54.95 | 54.00 |
| DeepSeek-Coder | BiGRU | $1e^{-5}$ | AdamW | 256 | 54.98 | 55.01 | 54.97 |

## E  COMMONSENSE REASONING

We evaluate various encoder-only and decoder-only LLM architectures augmented with RNN variants to assess the effectiveness of RNN-coupled LLMs on commonsense reasoning task across three publicly available datasets. Comparative performance in terms of accuracy (**A**) and F1 score is illustrated in Figure 4 and Table 1, while detailed experimental results are provided in Table 6. Notably, the LLM-RNN models consistently achieve higher $\mathbf{F1}\psi$, $\mathbf{P}\psi$, and $\mathbf{R}_\psi$ scores than their base LLM counterparts. However, on the Twitter US Airline dataset, performance gains were limited compared to the IMDb and Sentiment140 datasets. This is primarily attributed to severe class imbalance—62.69% of tweets are labeled as negative, 16.14% as positive, and 21.17% as neutral.

To address class imbalance in the Twitter dataset, we performed data augmentation by synthetically generating samples for the neutral and positive classes to match the number of negative samples. The augmented dataset was then used to train the RoBERTa-BiLSTM model, as illustrated in Figure 10. Following augmentation, the model demonstrated substantial performance gains, achieving $\mathbf{F1}_\psi$ and **A** scores of 95.74% and 95.77%, respectively—an improvement of approximately 15% ($\uparrow$) over its performance on the original imbalanced dataset (see in Table 6).

## F  CODE UNDERSTANDING

We conduct experiments using publicly available benchmark datasets for code defect detection, along with real-world code datasets, to evaluate and compare model performance on code understanding tasks. The best performance scores of both the base LLMs and their RNN-coupled variants on the

Table 8: Performance comparison in terms of weighted $\mathbf{F1}\psi$, $\mathbf{P}\psi$, and $\mathbf{R}_\psi$ on real-world datasets, using encoder-only, decoder-only, and encoder-decoder models coupled with RNNs and their base models. **Bold** values indicate performance improvement.

| LLM | RNN | SearchAlg | | | SearchSortAlg | | | SearchSortGTAlg | | |
|---|---|---|---|---|---|---|---|---|---|---|
| | | $\mathbf{F1}_\psi$ | $\mathbf{P}_\psi$ | $\mathbf{R}_\psi$ | $\mathbf{F1}_\psi$ | $\mathbf{P}_\psi$ | $\mathbf{R}_\psi$ | $\mathbf{F1}_\psi$ | $\mathbf{P}_\psi$ | $\mathbf{R}_\psi$ |
| RoBERTa | - | 90.53 | 90.55 | 90.54 | 94.40 | 94.51 | 94.55 | 92.79 | 92.89 | 92.92 |
| | LSTM | 93.48 | 93.67 | 93.44 | 95.62 | 95.72 | 95.69 | 95.19 | 95.51 | 95.19 |
| | BiLSTM | 93.59 | 93.72 | 93.55 | **96.25** | 96.25 | 96.26 | 95.34 | 95.46 | 95.36 |
| | GRU | 93.47 | 93.59 | 93.44 | 95.96 | 96.01 | 96.00 | 95.42 | 95.54 | 95.45 |
| | BiGRU | **93.63** | 93.90 | 93.59 | 95.99 | 96.02 | 96.00 | **96.00** | 96.10 | 96.00 |
| CodeBERT | - | 92.36 | 92.40 | 92.33 | 94.73 | 94.93 | 94.80 | 93.22 | 93.37 | 93.36 |
| | LSTM | 93.63 | 93.73 | 93.59 | 96.19 | 96.22 | 96.17 | **96.04** | 96.21 | 96.02 |
| | BiLSTM | **94.04** | 94.11 | 94.01 | **96.34** | 96.37 | 96.33 | 95.75 | 95.86 | 95.75 |
| | GRU | 93.85 | 93.91 | 93.82 | 96.09 | 96.11 | 96.10 | 96.04 | 96.13 | 96.05 |
| | BiGRU | 94.01 | 94.15 | 93.97 | 96.28 | 96.29 | 96.30 | 95.71 | 95.83 | 95.75 |
| CodeT5 | - | 93.63 | 93.67 | 93.61 | 94.88 | 95.14 | 94.88 | 96.01 | 96.03 | 96.15 |
| | LSTM | 94.40 | 94.41 | 94.40 | **96.72** | 96.76 | 96.72 | 95.98 | 96.11 | 96.07 |
| | BiLSTM | **95.12** | 95.15 | 95.12 | 96.68 | 96.68 | 96.68 | **96.01** | 96.06 | 96.07 |
| | GRU | 94.17 | 94.25 | 94.17 | 96.31 | 96.43 | 96.31 | 95.25 | 95.26 | 95.44 |
| | BiGRU | 94.86 | 94.86 | 94.86 | 96.28 | 96.29 | 96.32 | 94.98 | 95.15 | 95.06 |
| CodeT5$^+$ | - | 93.38 | 93.39 | 93.38 | 96.42 | 96.44 | 96.43 | 96.26 | 96.32 | 96.28 |
| | LSTM | 94.00 | 94.02 | 93.99 | 95.97 | 96.05 | 95.97 | 96.05 | 96.17 | 96.08 |
| | BiLSTM | 94.26 | 94.28 | 94.24 | 96.36 | 96.37 | 96.37 | **96.26** | 96.31 | 96.27 |
| | GRU | 94.27 | 94.30 | 94.26 | **96.42** | 96.44 | 96.42 | 95.92 | 96.06 | 95.92 |
| | BiGRU | **94.42** | 94.43 | 94.42 | 96.33 | 96.39 | 96.34 | 96.03 | 96.15 | 96.03 |
| DeepSeek-Coder | - | 94.79 | 94.81 | 94.78 | 97.23 | 97.23 | 97.23 | 94.78 | 94.84 | 94.87 |
| | LSTM | 94.27 | 94.31 | 94.24 | 96.77 | 96.78 | 96.78 | 92.97 | 92.90 | 92.86 |
| | BiLSTM | 94.03 | 94.07 | 94.01 | 96.69 | 96.70 | 96.71 | 92.97 | 93.08 | 93.03 |
| | GRU | 94.42 | 94.50 | 94.40 | 96.56 | 96.56 | 96.58 | 93.37 | 93.57 | 93.47 |
| | BiGRU | 93.96 | 94.00 | 93.94 | 96.77 | 96.77 | 96.78 | 92.91 | 93.04 | 92.97 |

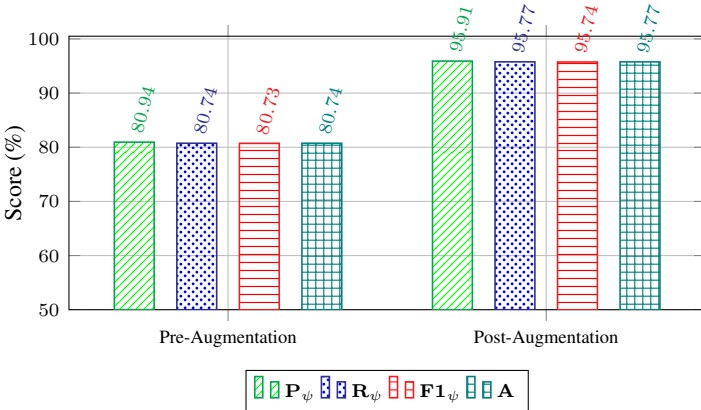

Figure 10: Performance comparison of the RoBERTa-BiLSTM model on the Twitter dataset pre- and post- data augmentation.

defect detection benchmark are reported in Table 2, with extended comparative results provided in Table 7. Furthermore, we evaluate our approach on three real-world datasets—SearchAlg, Search-SortAlg, and SearchSortGTAlg—with results summarized in Table 3 and visualized in Figure 5. Across all datasets, we observe that integrating RNNs with encoder-only, and encoder-decoder LLMs consistently enhances performance over their respective base models. Table 8 presents a comprehensive evaluation on real-world datasets, reporting the performance of encoder-only, decoder-only, and encoder-decoder LLMs, both in their base forms and when coupled with various RNN variants. The results are shown in terms of weighted $\mathbf{F1}\psi$, $\mathbf{P}\psi$, and $\mathbf{R}\psi$ scores. Notably, the RNN-coupled encoder-only and encoder-decoder LLMs consistently outperform their standalone counterparts in

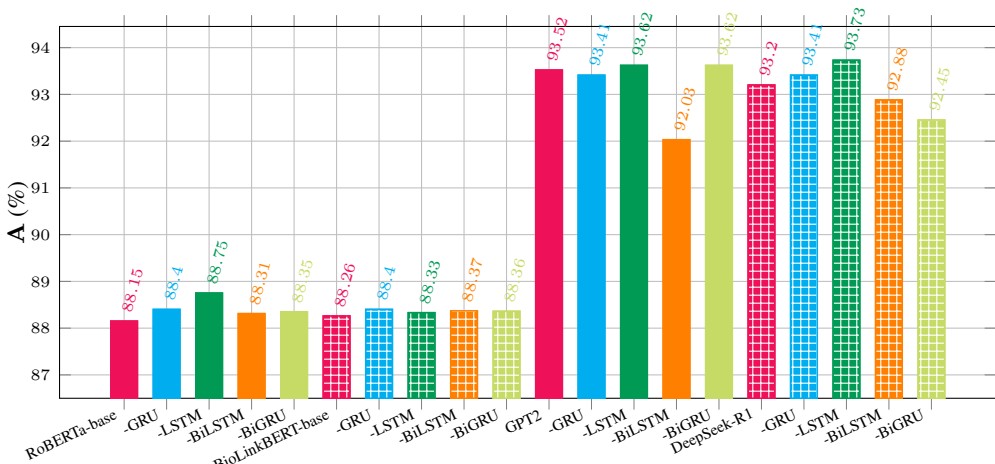

Figure 11: Comparison of the accuracy scores of RoBERTa-base, BioLinkBERT-base, GPT2, and DeepSeek-R1 models, along with their RNN variants, on the NCBI dataset. The decoder-only models (GPT2 and DeepSeek-R1) achieved higher accuracy than the encoder-only models (RoBERTa and BioLinkBERT).

$\mathbf{F1}\psi$ score, indicating a clear performance gain. In contrast, RNN-coupled decoder-only LLMs fail to exhibit improvements over their base versions. These findings provide strong evidence that coupling RNNs with LLMs can enhance code understanding capabilities.

# G  BIOMEDICAL REASONING

The highest accuracy achieved by the RNN-coupled LLMs and their corresponding base models is depicted in Figure 6, with extended comparative results provided in Figure 11. The best scores for $\mathbf{F1}\psi$, $\mathbf{P}\psi$, and $\mathbf{R}_\psi$ are reported in Table 4, while the detailed results are presented in Table 9. Notably, decoder-only architectures demonstrate substantial performance improvements over encoder-only counterparts. Furthermore, in most cases, the RNN-coupled models outperform their standalone model in terms of both $\mathbf{A}$ and F1 score.

Table 9: Quantitative results for biomedical reasoning using RoBERTa, BioLinkBERT, GPT2, and DeepSeek-R1 models with RNNs and their base models on the NCBI dataset.

| Model | $\mathbf{F1}_\psi$ | $\mathbf{P}_\psi$ | $\mathbf{R}_\psi$ |
|---|---|---|---|
| RoBERTa-Base | 86.75 | 85.40 | 88.15 |
| -GRU | 86.85 | 85.36 | 88.40 |
| -LSTM | 87.02 (↑) | 85.37 | 88.75 |
| -BiGRU | 86.83 | 85.38 | 88.35 |
| -BiLSTM | 86.80 | 85.36 | 88.31 |
| BioLinkBERT-Base | 86.81 | 85.43 | 88.26 |
| -GRU | 86.86 (↑) | 85.39 | 88.40 |
| -LSTM | 86.84 | 85.42 | 88.33 |
| -BiGRU | 86.82 | 85.33 | 88.37 |
| -BiLSTM | 86.84 | 85.39 | 88.36 |
| GPT2 | 93.52 | 93.52 | 93.52 |
| -GRU | 93.40 | 93.41 | 93.41 |
| -LSTM | 93.63 (↑) | 93.64 | 93.62 |
| -BiGRU | 93.62 | 93.62 | 93.62 |
| -BiLSTM | 92.05 | 92.15 | 92.03 |
| DeepSeek-R1 | 93.20 | 93.20 | 93.20 |
| -GRU | 93.43 | 93.53 | 93.41 |
| -LSTM | 93.75 (↑) | 93.98 | 93.73 |
| -BiGRU | 92.48 | 92.59 | 92.45 |
| -BiLSTM | 92.91 | 93.23 | 92.88 |

## H ABLATION STUDY

We conducted an additional ablation study examining how LLM-RNN performance changes when the RNN is attached to early, intermediate, or final layer embeddings of the LLM. Table 10 shows the results for GPT2 on commonsense datasets. We observed that attaching the RNN to the final layer embeddings consistently yields the best performance. For example, GPT2 (final layer) + GRU achieves approximately 6.84% and 4.36% higher F1 scores compared to attaching the RNN to the early and intermediate layers, respectively.

Table 10: Ablation study with different embedding layers of GPT2 and RNNs

| Model | Dataset | $F1_\psi$ (early layer) | $F1_\psi$ (inter. layer) | $F1_\psi$ (final layer) |
|---|---|---|---|---|
| GPT2-LSTM | Twitter | 76.31 | 78.34 | 78.86 |
| GPT2-GRU | IMDB | 85.08 | 87.56 | 91.92 |
| GPT2-BiLSTM | Sentiment140 | 78.98 | 80.76 | 80.97 |

