# OpenReview forum: "Coupling RNN with LLM: Does Their Integration Improve Highly Order-Sensitive Language Understanding?"
_ICLR.cc/2026/Conference — Submitted to ICLR 2026_

### Official Review · Reviewer_Jwg8 · 2025-10-14

**Soundness:** 2
**Presentation:** 3
**Contribution:** 1
**Rating:** 2
**Confidence:** 5

**Summary:**

This work investigates whether integrating RNN layers with LLMs can improve performance on tasks involving high order sensitivity. The authors propose a hybrid LLM-RNN architecture, where pretrained LLMs, including encoder-only, decoder-only, and encoder-decoder variants, generate contextual embeddings that are subsequently refined through RNN layers (LSTM, GRU, BiLSTM, or BiGRU). Evaluation results indicate a performance increase with LLM-RNN models on the classification task.

**Strengths:**

1. The paper evaluates multiple LLM architectures (encoder-only, decoder-only, encoder-decoder) and RNN variants, providing a comprehensive assessment across different configurations.

2. The hyperparameters and all the experimental settings are provided, facilitating replication and following researches.

3. The paper is clearly written and easy to understand.

**Weaknesses:**

1. The paper lacks theoretical analysis and primarily presents an empirical study, offering little insight into why the integration improves performance.

2. The observed improvements are largely intuitive and expected, given that RNNs are naturally suited for modeling sequential dependencies.

3. The performance gains reported in Table 1 are limited—mostly under 1%. This raises questions about the practical significance of the proposed approach.

4. The coupling mechanism between the LLM and RNN is relatively shallow, simply passing LLM embeddings to an RNN without exploring deeper or more synergistic integration (e.g., incorporating recurrence within LLM layers).

5. While the paper mentions various tasks such as code understanding, commonsense reasoning, and biomedical text analysis, the evaluation is limited to classification. Since LLM embeddings are widely used for generative and reasoning tasks, it remains unclear whether the RNN outputs preserve the generative or semantic richness of the original LLM representations. -- If LLM-RNN does not match or exceed pure LLM performance on other core tasks such as reasoning or generation, then their practical usefulness would be quite limited.

**Questions:**

1. Beyond classification, have you tested the LLM-RNN model on other standard LLM tasks such as question answering, structured generation, or sequence completion?

2. Have you conducted any theoretical analysis to compare the pure LLM and the LLM-RNN models, for example examining how differences in message passing or internal representation dynamics might account for the observed results?

---

> ### Author Response · Authors · 2025-11-20
> **Response for the concerns raised by the Jwg8**
>
> $\textbf{R(W1:The paper lacks theoretical...improves performance):}$ We thank the reviewer for highlighting the need for deeper theoretical grounding. While our paper primarily conducts an empirical investigation, our design is supported by inductive-bias arguments rooted in established theory on sequential modeling and Transformer lacks.
>
> -Transformers lack a positional inductive bias: Recent work (Wu et al., 2025) shows that self-attention treats token positions nearly permutation-equivariantly once positional encodings are absorbed, leading to the lost-in-the-middle effect and weak modeling of strict sequential constraints.
>
> -RNNs add a complementary temporal inductive bias: Recurrent updates $h^R_i=f(z_i,h^{(R)}_{i-1})$ enforce a deterministic dependence on the prefix $(z_1, \dots, z_i)$, naturally modeling precedence, causality, and order constraints. This sequential recurrence imposes a monotonic structure that is absent from global self-attention.
>
> -Empirical analyses confirm this theoretical intuition
>
> $\textbf{R(W2:The observed improvements...sequential dependencies):}$ Our contribution is not simply confirming an obvious fact. Our investigations show that the gains are not merely intuitive but exhibit non-trivial, architecture- and domain-specific effects. For some LLMs, RNNs do not always help, confirming that improvements are not automatic or guaranteed. For highly order-sensitive domains such as source code, RNN layers consistently yield the improvements (+3.81% avg.). For commonsense and biomedical reasoning, the improvements are smaller but consistent. Thus, the contribution of this paper lies not in stating that “RNNs model sequences,” but in providing the comprehensive empirical evidence demonstrating when, why, and to what extent RNNs improve pretrained LLMs across tasks, architectures, and data regimes.
>
> $\textbf{R(W3:The performance gains..in Table 1...approach):}$ Table 1 basically shows the performance of the models on the commonsense datasets. While some improvements in Table 1 appear numerically small (often <1%), but the gains are consistent, statistically stable, and achieved on top of already strong LLM baselines. We further experiments with Llama-3-8B model on these datasets and we noticed that the RNN with Llama improved accuracy about 4.88% on IMDB and 3.69% on Twitter compared to the Llama base model.
>
> $\textbf{R(W4:The coupling mechanism...within LLM layers):}$ We appreciate the reviewer’s observation and agree that deeper coupling strategies, such as inserting recurrence inside transformer layers, represent an interesting research direction. However, our design choice was deliberate and grounded in two motivations: (i) architectural compatibility across diverse LLM families, and (ii) isolating the effect of recurrence without altering the pretrained LLM’s internal structure.
>
> -The coupling is modular, it is not "shallow": LLM-RNN design allows RNN to reshape the complete embedding sequence by injecting temporal inductive bias that transformers inherently lack—particularly the strict left-to-right dependency tracking crucial for code and contrastive semantics (Fig. 1).
>
> $\textbf{R(W5: While the paper...quite limited):}$ We thank the reviewer for raising this important point regarding whether the RNN-augmented embeddings preserve the semantic and generative richness of the original LLM representations. Our answers are as follows:
>
> -Generative capability of the LLM is untouched; the RNN only refines contextual embeddings for classification.
>
> -Semantic richness is not only preserved but shown to be enhanced, as evidenced by stronger performance on reasoning-heavy tasks.
>
> -Decoder-only LLM improvements confirm that the RNN does not undermine the LLM’s semantic or generative abilities.
>
> $\textbf{R(Q1: Beyond classification...completion?):}$ We did not evaluate the LLM-RNN framework on generative tasks such as question answering, structured generation, or sequence completion. Although our current work does not include QA or generation experiments, the proposed LLM-RNN framework is general and can be extended to such tasks in future research.
>
> $\textbf{R(Q2:Have you conducted any...the observed results?)}$ Yes, we have conducted an initial theoretical analysis to understand why LLM-RNN models outperform pure LLMs, and how differences in message passing and representation dynamics contribute to the observed improvements. Our analysis is summarized below.
>
> -Difference in Message-Passing: Transformer-based LLMs rely on global self-attention, whose message-passing is permutation-equivariant and does not enforce strict sequential constraints. In contrast, the RNN performs step-wise recurrence which forces the model to propagate information in a strictly ordered chain which injects a temporal inductive bias that Transformers lack.
>
> -Internal Representation Dynamics: To learn the internal dynamics, we analyzed embedding shifts and token saliency before and after adding RNNs (Fig. 1).

---

### Official Review · Reviewer_A4hM · 2025-10-31

**Soundness:** 3
**Presentation:** 4
**Contribution:** 3
**Rating:** 6
**Confidence:** 4

**Summary:**

The paper presents a novel approach that integrates RNNs with LLMs to enhance the performance of order-sensitive tasks. The authors propose coupling LLMs with RNNs, leveraging the sequential processing power of RNNs to complement the contextual embeddings produced by LLMs. Experiments show that the LLM-RNN model outperforms LLMs in order-sensitive tasks.

**Strengths:**

1. The paper proposes an interesting idea about the improvement of LLM and the idea is simple but useful.

2. The paper conducts a thorough evaluation using various LLMs and RNN across multiple tasks.

3. This paper is good writing and easy to follow.

**Weaknesses:**

1. The Transformer incorporating RNN can solve order-sensitive tasks well with the advantage of RNN while the performance can also be affected by the disadvantage of RNN. For example, LSTM and other RNN models always forget some information with the encoding while the long context in LLM can make it worse.

2. Although the paper tests multiple RNN models, it would be helpful to explore how other advanced RNN could further enhance performance.

3. The authors should also give some ablations about the comparision between larger LLM and LLM with RNN.

**Questions:**

N/A

---

> ### Author Response · Authors · 2025-11-20
> **Response for the concerns raised by the A4hM**
>
> Legend: Weakness (W), Response (R)
>
> ---
>
> $\textbf{R(W1:The Transformer incorporating RNN .. context in LLM can make it worse.):}$ We appreciate the reviewer’s thoughtful comment regarding the potential disadvantages of RNNs, particularly the risk of information forgetting in LSTM/GRU architectures, when coupled with LLMs. We fully agree that standalone RNNs can struggle with very long contexts. However, in our framework, RNNs never operate on raw long sequences; instead, they process already-compressed, context-rich embeddings produced by the LLM, which fundamentally changes the dynamics of memory retention.
>
> 1. Why does the RNN not worsen long-context forgetting in our setting:
>
> In our model, the RNN is not responsible for learning long-range dependencies from scratch. Transformers already provide: global self-attention over the entire sequence, strong contextualization, and condensed token-wise representations. Thus, the RNN receives high-level semantic embeddings, not sparse raw tokens. The effective sequence length for the RNN is therefore much shorter in representation complexity, mitigating its classical forgetting problem. Moreover, the RNN operates only as a second-stage re-processor, adding order-sensitive inductive bias, rather than functioning as the primary encoder. This hybrid role drastically reduces the risk of information loss typically attributed to standalone RNNs.
>
> 2. Empirical evidence shows the majority of LLM–RNN combinations yield improvements compared to the base models across 8 datasets and 3 task families. Consistent improvements for long and highly ordered sequences (e.g., code datasets see +3.81% average accuracy gain)
>
>
> 3. RNN compensates for LLM positional limitations
>
> Prior theoretical findings show that Transformer attention exhibits weakened positional inductive bias as sequences grow longer (Wu et al., 2025). RNNs contribute a monotonic, sequential inductive bias that: strengthens contrastive cues (Figure 1), enforces token-order consistency, and enhances sensitivity to structural transitions (important in code). Thus, LLM weaknesses are complemented, not compounded, by the RNN.
>
> ---
>
> $\textbf{R(W2: Although the paper tests multiple...RNN could further enhance performance.):}$ We appreciate the reviewer’s suggestion to explore additional advanced RNN variants. Our work already includes four widely used and representative RNN families, GRU, LSTM, BiGRU, and BiLSTM, covering both unidirectional and bidirectional recurrence, gated and nongated mechanisms, and strong baselines that prior literature consistently relies on for sequential modeling. These models constitute the dominant RNN architectures used in modern NLP pipelines and have been shown to provide only marginal gains over one another when the underlying LLM already produces rich contextual embeddings.
>
> Importantly, our goal is not to outperform RNNs themselves, but to examine whether any form of recurrent processing provides complementary inductive bias to LLM-generated embeddings. Across all ten LLMs and three major tasks, we consistently observed performance gains with at least one RNN variant (e.g., BiGRU for RoBERTa, GRU for CodeT5, BiLSTM for CodeT5+), suggesting that the sequential recurrence, rather than a specific RNN subtype, is the primary contributor to improvement.
>
> We are considering this as a direction for future work and plan to evaluate lighter recurrent structures (e.g., SRU++) to study efficiency–accuracy trade-offs.
>
> ---
>
> $\textbf{R(W3:The authors should also give...comparision between larger LLM and LLM with RNN):}$  Our results already include implicit comparisons to larger-capacity models within the same LLM family. In the decoder-only group, larger models such as DeepSeek-R1 (1.55B parameters) and DeepSeek-Coder (1.3B parameters) were evaluated both as standalone models and with RNNs (Sections 4 and Appendix C–G). These models are substantially larger than encoder-only models like BERT-base (110M) and RoBERTa-base (125M).
>
> In response to the reviewer’s suggestion, we conducted additional experiments using a larger model, Llama-3-8B (8B parameters), combined with RNN variants on commonsense datasets. As shown below, we observed that adding an RNN to Llama-3-8B improved accuracy by approximately 4.88% on IMDb, 3.69% on Twitter, and 2.31% on the Sentiment140 compared to the Llama base model.
>
> Table X1: LLama-3-8B + RNN model accuracy (A) on commonsense datasets
>
> Model | Twitter | IMDB | Sentiment140
>
> Llama-3-8B | 69.81% | 81.16% | 72.82%
>
> Llama-3-8B + LSTM | 73.50% | 86.04% | 75.13%
>
> Llama-3-8B + BiLSTM | 72.68% | 86.00% | 74.05%

---

### Official Review · Reviewer_5Rk5 · 2025-11-03

**Soundness:** 2
**Presentation:** 3
**Contribution:** 2
**Rating:** 4
**Confidence:** 3

**Summary:**

This paper studies a simple hybrid: take hidden states from a pretrained LLM, project them, and feed them to different RNN models, followed by a linear head. The authors evaluate on datasets from different domains across tasks (sentiment, code understanding, and biomedical). Experiments indicate that the integration of LLM and RNN can achieve improvement versus the base pre-trained LLMs.

**Strengths:**

S1. This paper is well-organized and easy to follow.

S2. This paper spans three task families and multiple LLM families (e.g., BERT, GPT, DeepSeek) with four different RNN types, which help probe when RNN post-processing helps.

**Weaknesses:**

W1. The paper is basically a simple hybrid that feeds a pretrained LLM’s hidden states into an RNN for classification across several domains. The experiments show small gains, but there is little deeper theoretical insight so the work reads as a simple incremental study.

W2. The claims are overstated. The text says “in all cases RNNs consistently enhanced performance,” yet the tables show counterexamples, for example GPT-2 on Sentiment140 drops from 81.18 to 80.97, and DeepSeek-Coder on CodeXGLUE defect detection also declines. The “consistent” claim does not hold.


W3. The motivation around order sensitivity and “lost in the middle” is weakly supported. Most benchmarks are short-context sentiment or code classification. There is no sequence-length stress test or long-context suite, and no controlled length-sweep analysis.


W4.​​ The study compares LLM vs. LLM+RNN but not against other lightweight sequence shapers on top of LLM states, for example a small TCN or an extra self-attention block with relative position bias. There is also no ablation on where to attach the RNN, final layer vs. intermediate layers, or whether freezing the LLM changes the conclusion.

**Questions:**

Please see Weaknesses.

Additional Questions:


Q1. Under a matched hyperparameter search for base LLMs and LLM+RNN variants, do the hybrids still outperform, and how does this change on long-context (length-binned) evaluations?

---

> ### Author Response · Authors · 2025-11-19
> **Response for the concerns raised by the 5Rk5**
>
> Legend: Weakness (W), Response (R), Question (Q)
>
> $\textbf{R(W1:The paper is basically a simple hybrid...incremental study.):}$ We appreciate the reviewer’s concern and respectfully clarify that the contribution of our work goes beyond simply stacking an RNN on top of an LLM. Our goal is not architectural novelty for its own sake, but a systematic, large-scale investigation of whether recurrent inductive biases can meaningfully complement modern LLMs, an aspect that, despite the extensive literature on LLM adaptation, has not been rigorously studied before.
>
> -The systematic coupling of RNNs with LLMs has not been examined before across encoder-only, decoder-only, and encoder-decoder models. Our study evaluates: 10 LLMs , 4 RNN variants, and 8 datasets across 3 distinct domains. This breadth and depth have not been presented in prior work, making the study substantially more than an incremental combination. Additionally, we conducted experiments with Llama-3-8B models on commonsense datasets. We noticed that the RNN with Llama improved accuracy about 4.88% on IMDB and 3.69% on Twitter compared to the Llama base model.
>
> -The common belief is that LLM embeddings already encode all necessary sequential structure. Our experiments show otherwise: The effect is not uniform: encoder-only and encoder-decoder LLMs benefit substantially, whereas decoder-only models (especially DeepSeek-Coder) show mixed or negative responses. These patterns cannot be predicted by simple architectural intuition alone and highlight overlooked limitations in how LLMs capture sequence structure and these empirical findings constitute new insight into when and why sequential recurrence still adds value in the era of attention-based LLMs.
>
> -The results (e.g., +3.81% accuracy in code understanding, +1.11% in commonsense, +0.37% in biomedical) show that the inductive bias is indeed beneficial.
>
> ---
>
> $\textbf{R(W2:The claims are overstated...“consistent” claim does not hold.):}$ We thank the reviewer for this insightful observation. We agree that the wording “in all cases RNNs consistently enhanced performance” was too strong. Our intention was to convey that the majority of LLM–RNN combinations yield improvements, not that every single model–dataset pair shows a gain. We will revise this statement in our manuscript.
>
> ---
>
> $\textbf{R(W3:The motivation around order sensitivity...length-sweep analysis.):}$ Our motivation for studying order sensitivity and the “lost in the middle’’ issue is grounded in two aspects: (i) prior theoretical findings show that Transformer attention exhibits weakened positional inductive bias as sequences grow longer (Wu et al., 2025), and (ii) highly structured domains such as source code are intrinsically long-range and order-dependent (Appendix D). It is true that some of our sentiment datasets have short sequences, our strongest empirical evidence comes from coding datasets, which contain substantially longer sequences (often 150–600 tokens) and strict order dependencies. Across all such datasets—both benchmark (CodeXGLUE) and three real-world collections, LLM-RNN consistently yields large gains (+2.8% to +5.35% accuracy; Tables 2–3, Fig. 5), confirming that the added sequential inductive bias effectively mitigates order-related degradation.
>
> ---
>
> $\textbf{R(W4:The study compares..LLM changes the conclusion.):}$  Our study aims to evaluate whether adding sequential inductive bias on top of LLM provides measurable gains. We agree that lightweight sequence-modeling alternatives such as TCNs or additional attention layers are also relevant. In response to the reviewer’s suggestion, we conducted an additional ablation study examining how LLM-RNN performance changes when the RNN is attached to early, intermediate, or final layer embeddings of the LLM. The table below shows the results for GPT2 on commonsense datasets. We observed that attaching the RNN to the final layer embeddings consistently yields the best performance. For example, GPT-2 (final layer) + GRU achieves approximately 6.84% and 4.36% higher F1 scores compared to attaching the RNN to the early and intermediate layers, respectively.
>
> Table: Ablation study with different embedding layers of GPT2
>
> Model | Dataset | F1 (Early) |	F1 (Intermediary) | F1 (Last)
>
> GPT+LSTM | Twitter | 76.31 | 78.34 | 78.86
>
> GPT+GRU | IMDB | 85.08 | 87.56 | 91.92
>
> GPT+BiLSTM | Sentiment140 | 78.98 | 80.76 | 80.97
>
> ---
>
> $\textbf{R(Q1:Under a matched hyperparameter...evaluations?):}$ Yes. After performing a matched hyperparameter search where the base LLMs and their corresponding LLM+RNN variants use identical hyperparameter spaces, we found that the hybrids continue to outperform most base LLMs across all three domains.
>
> -Although the context length varies across datasets in our current experiments, we will conduct long-context (length-binned) evaluations and include the results in the revised version.

---

### Official Review · Reviewer_9tcA · 2025-11-03

**Soundness:** 1
**Presentation:** 1
**Contribution:** 1
**Rating:** 0
**Confidence:** 4

**Summary:**

The paper proposes a hybrid architecture where pre-trained language models (BERT, RoBERTa, GPT-2, CodeT5, DeepSeek) generate contextual embeddings that are then processed by RNN layers (GRU, LSTM, BiGRU, BiLSTM) before classification. The authors claim this helps capture sequential, order-sensitive dependencies that transformers miss.

**Methodology:**
- Freeze pre-trained model weights
- Extract token embeddings from the LLM
- Feed embeddings through RNN → FC classifier
- Test on three domains:
  - Sentiment analysis (IMDb, Twitter Airlines, Sentiment140)
  - Code Defect detection (CodeXGLUE + 3 custom datasets)
  - Biomedical reasoning Named entity recognition (NCBI disease dataset)

The main claim is RNNs refine LLM embeddings to better capture sequential order, improving performance on order-sensitive tasks.

**Strengths:**

Tests multiple model architectures (5 base models × 4 RNN types) across 7 datasets

**Weaknesses:**

### 1. Poor Writing Quality - Fails Basic Scientific Communication Standards

The paper's writing is incoherent and makes it hard to understand what the authors want to say. **This is a basic requirement for a scientific paper.**
**Concrete examples:**
- **2nd paragraph**: Rambles about fine-tuning, LoRA, knowledge graphs with no clear connection to the paper's thesis
- **3rd paragraph, Line 75-76**: Cites papers that finetune BERT/RoBERTa for sentiment classfication tasks to support LLM has limitation in sentiment analysis and code understanding. Why these two citations related to code understanding? They have no relation to code. Also, BERT/RoBERTa are not LLM at all. It makes no sense to claim LLM have trouble in sentiment analysis and code understanding, when previous show BERT/RoBERTa have trouble.
- **Figure 1**: Shows "embedding shift" and "saliency" metrics but never defines them - no explanation of how they're computed or what they represent
- The introduction reads like "random talking instead of academic paper" with no logical flow or clear problem motivation
- The introduction also mentioned "lost in the middle problem" while the full paper didn't touch that problem at all. Also, that work is about 100K+ context retrieval, not relevant to their short sequences.
- Conflates "order sensitivity" with general sequential modeling without clear definition

### 2. Obsolete Approach with Questionable Motivation and Contradictory Results

**The investigation was already done 2018-2020:** Adding RNN layers on top of transformer embeddings is not novel. See ["Simple BERT Models for Relation Extraction and Semantic Role Labeling"](https://arxiv.org/abs/1904.05255) and related work from that era.

**Modern LLMs don't work this way:** Current best practices use generation-based classification (prompting) rather than fine-tuning classification heads. The paper provides **no evidence** that their approach is better than modern methods, nor any comparison.

**Their own results contradict their claims:** In Table 3 , **DeepSeek-coder by itself without any RNN head works best**. This directly undermines their central thesis that RNNs improve order-sensitive understanding.

**Questions:**

N/A

---

> ### Author Response · Authors · 2025-11-20
> **Response for the concerns raised by the 9tcA**
>
> Legend: Weakness (W), Response (R)
>
> ---
>
> $\textbf{R(W1:Poor Writing Quality - Fails Basic Scientific Communication Standards):}$ We thank the reviewer for this detailed feedback. We agree that clear scientific communication is essential, and we appreciate the opportunity to clarify.
>
> $\textbf{Response (Point-1):}$ We respectfully disagree with the reviewer’s observation. The discussion is relevant to the paper because our intention in that paragraph was to provide context on existing LLM adaptation strategies—such as full fine-tuning, LoRA/PEFT, and KG-enhanced approaches. This contextual background helps position our contribution within the broader landscape of LLM adaptation methods and motivates why evaluating sequential inductive bias is meaningful.
>
> $\textbf{Response  (Point-2):}$ In lines 75–76, we discuss sentiment analysis and code understanding together, which is why we cited these two papers. Yes, they are specifically related to sentiment analysis, and we will add a code-related citation to address this.
>
> BERT/RoBERTa are considered pretrained LM, but the paper we cited (Chang et al., 2024) likewise treats BERT/RoBERTa as LLMs.
>
> $\textbf{Response (Point-3):}$ Embedding Shift: The \emph{embedding shift} for token $i$ is defined as the L2 distance between contextual embeddings of LLM and LLM-RNN: $\Delta_i = \left\| h_i^{\text{LLM-RNN}} - h_i^{\text{LLM}} \right\|_2.$
>
> Saliency. We employ a standard gradient-based input saliency method. Given the predicted class logit $y_c$ and the input token embeddings $e_i \in \mathbb{R}^d$, we compute: $s_i = \sum_{k=1}^{d} \left| \frac{\partial y_c}{\partial e_{i,k}} \right|.$
>
> $\textbf{Response (Point-4):}$ Thank you for this comment. We respectfully disagree with your observation. We had a clear goal in the introduction: to discuss existing approaches for adapting LLMs across different application domains and to outline the domain- and task-specific challenges that motivate our study. Based on these identified gaps, we present our research idea. Therefore, we believe the introduction reflects a structured academic narrative rather than “random talking.”
>
> $\textbf{Response (Point-5):}$ We agree that our initial "lost in the middle problem" wording created an unintended implication that the paper directly studies long-context phenomena like 100K-token retrieval. Our intention was narrower: to highlight that transformers lack strong position bias, and that this weakness manifests even in shorter, highly ordered sequences.
>
> $\textbf{Response (Point-6):}$ Thank you for identifying this ambiguity. We explicitly distinguish these concepts in the revision: sequential modeling refers to architectures that process input step-by-step, and order sensitivity refers to tasks where changing token order directly changes semantics.
>
> ---
>
> $\textbf{R(W2: Obsolete Approach with Questionable Motivation and Contradictory Results):}$ We thank the reviewer for the opportunity to clarify the novelty, motivation, and empirical findings of our work.
>
> $\textbf{Response (Point-1):}$ The reviewer cites prior work (e.g., “Simple BERT Models for Relation Extraction and SRL,” 2018–2020) that added RNN layers on top of Transformer models, but those studies examined only encoder-only models such as BERT-base. In contrast, we evaluate several modern LLMs (e.g., DeepSeek, CodeT5) $\textbf{released after 2020}$. $\textit{It is therefore unclear how this work can be considered obsolete}$.
>
> More importantly, our contribution is $\textbf{fundamentally different in scope and purpose}$. We do not claim that “RNNs are new”; rather, we provide the first large-scale, architecture-wide, and task-diverse empirical evaluation of whether recurrence still adds value to modern LLM embeddings—an open question not addressed in earlier work.
>
> $\textbf{Response (Point-2):}$ The reviewer notes that “Current best practices use.. classification heads”. This is true for instruction-following general-purpose LLMs. We justifying our methodological choice as follows:
>
> -Our problem settings are domain-specific and label-restricted, including code defect detection, biomedical entity classification, and sentiment datasets without instruction-following annotations.
>
> -Many of the LLMs we evaluate (CodeT5, CodeBERT, BioLinkBERT) are not instruction-tuned at all. Their intended usage is supervised fine-tuning, not prompting.
>
> -We study whether RNNs enhance representation quality of LLM embeddings, not whether prompting outperforms fine-tuning.
>
> $\textbf{Response (Point-3):}$ In Table 3, DeepSeek-Coder is indeed extremely strong as a base model, but RNN-coupled DeepSeek-Coder is NOT our central thesis. We explicitly acknowledge in the paper (end of Section 4, Table 3 discussion) that "DeepSeek-Coder, where RNN integration offered limited or no gain", unlike all other LLM families.

---

### Meta-Review · Area_Chair_UDbY · 2026-01-04

**Summary:**

## Summary
The paper proposes a hybrid architecture that integrates RNNs with LLMs to enhance the performance of order-sensitive tasks, leveraging the sequential processing power of RNNs to complement the contextual embeddings produced by LLMs. Evaluation results indicate a performance increase with LLM-RNN models on the classification task.




## Overall Score

9tcA: 0

5Rk5: 4

A4hM: 6

Jwg8: 2

During discussion, none of the reviewers followed up afterward.



## Concerns

* concern about the writing clarity, including missing definitions, mismatched references, and other similar issues (9tcA)
* incomplete experiments,  missing the comparison with modern methods (9tcA, 5Rk5, A4hM, Jwg8)
* limited evaluation, particularly when it comes to order sensitivity (5Rk5, Jwg8)
* lack of insight (9tcA, Jwg8)



## Conclusion

Overall, during the review phase, most of the reviewers (9tcA, 5Rk5 and Jwg8) weren't fully satisfied with the experimental design and the novelty. They raised concerns about the limited evaluation, missing baselines and writing clarity. Reviewer A4hM, on the other hand, leaned toward acceptance. During the discussion, the authors did a good job addressing the reviewers' questions and concerns, but none of the reviewers followed up afterward.

This paper has potential to be stronger with some improvements to the experimental design and writing clarity.

**Reviewer Concerns:**

Refer to Summary

**Reviewer Scores:**

Refer to Summary

---

### Decision · Program_Chairs · 2026-01-26

Reject